# The Influence of Cognitive and Emotional Factors on Social Media Users’ Information-Sharing Behaviours during Crises: The Moderating Role of the Construal Level and the Mediating Role of the Emotional Response

**DOI:** 10.3390/bs14060495

**Published:** 2024-06-12

**Authors:** Yanxia Lu

**Affiliations:** School of Management, Liaoning Normal University, Dalian 116029, China; luyanxia@lnnu.edu.cn

**Keywords:** social media, information-sharing behaviour, heuristic–systematic model, cognitive appraisal, emotional response, construal level

## Abstract

Understanding the intricate dynamics of social media users’ information-sharing behaviours during crises is essential for effective public opinion management. While various scholarly efforts have attempted to uncover the factors influencing information sharing through different lenses, the underlying mechanisms remain elusive. Building upon the heuristic–systematic model (HSM) and construal level theory (CLT), this study explores the complex mechanisms that govern social media users’ information-sharing behaviours. The results indicate that both cognition and emotion play crucial roles in shaping users’ information-sharing behaviours, with systematic cues having the most significant impact on information-sharing behaviours. In terms of heuristic cues, positive emotions are more influential on information-sharing behaviours than primary cognition and negative emotions. Furthermore, spatial distance emerges as a key moderator, influencing individuals’ levels of engagement in information sharing. Emotion also acts as a mediator, connecting cognition to information sharing. This study provides insights into the sophisticated mechanisms of information sharing during crises, offering valuable implications for emergency management agencies to utilise social media for targeted public opinion guidance.

## 1. Introduction

In crises such as earthquakes, floods, fires, and other emergencies, due to the urgency of time, the severity of events, the unpredictability of the consequences, and other characteristics, people have different psychological processes [1]. Social media such as Weibo and WeChat provide people with more space for discourse and a place to express opinions and exchange views. The information content released through social platforms contains people’s attitudes and opinions, having an important impact on public psychology and information behaviours. In a crisis, people are very vulnerable to external stimuli, resulting in negative emotions such as fear and anxiety, which can lead to irrational information-sharing behaviours and endanger social stability once improperly controlled.

Scholars have conducted in-depth research on information-sharing behaviours. Previous research has developed the research model by involving variables that are derived from different theoretical perspectives such as the theory of reasoned action (TRA) [2], the theory of planned behaviour (TPB) [3], social cognitive theory [4,5], the uses and gratifications theory [6], the risk management theory, the protection motivation theory [7], the self-determination theory [8], the elaboration likelihood model (ELM) [9], etc. The fundamental basis of these theories usually pivots around two types of decision making. One is the reasoned action model, which states that behaviour stems from a rational decision making process, while the other is heuristic, implying that behaviour may ensue from emotional reactions to certain situations [10]. Attitudes and subjective norms are usually antecedents of behaviours in the reasoned path, while psychological images are usually included in the social reactive pathway to explain the behaviour.

Based on these theories, scholars have focused on the influence of information content [11], information source [12], context [13], and emotion [14]. Some scholars have also revealed the formation mechanism and behaviour rules of information sharing from a cross-cultural perspective [15]. Information content on social media can represent people’s inner activities to a certain extent, but the existing research rarely considers the psychological process of users, and it is difficult to effectively explore the root cause of information-sharing behaviours. 

During a crisis, individuals’ behavioural patterns change in response to alterations in their mental state, as external stimuli impact their cognitive appraisal and emotional response. The heuristic–systematic model (HSM) suggests that the impact of cognition and emotion on information behaviours can be understood through two distinct types of information processing: heuristic and systematic [16]. By integrating the HSM with empirical insights from emergency situations, this study seeks to elucidate the intricate psychological processes of cognitive appraisal and emotional response experienced by the public during crises. The goal is to unveil the underlying mechanisms of the information-sharing behaviours exhibited by users amidst crises, shedding light on this “black box” phenomenon.

Additionally, construal level theory (CLT) has been found to influence how people make and evaluate decisions [17], providing insights into the general principles governing crisis event judgment across different scenarios. This theory holds that the construal level influences an individual’s cognitive judgment [18], whereby individuals with a high construal level pay more attention to the abstract, essence, and de-contextualised content of the cognitive object, while individuals with a low construal level pay more attention to the concrete and context-dependent content of the cognitive object. Exploring information-sharing behaviours from the perspective of the construal level can reveal the differentiated impact of external stimuli on social media users. Although some studies have examined the role of the construal level, there is limited research on how the construal level specifically influences the information-sharing behaviours of social media users during crises.

To address this research gap and aim, a research framework for information-sharing behaviours in crises is conducted by integrating the HSM and CLT. Furthermore, the moderating role of the explanatory level and the mediating role of the emotional response are explored. To the best of my knowledge, this study represents the first empirical integration of HSM and CLT within the context of information-sharing behaviours during crises, paving the way for future research in this area of information behaviour. By extending the application of HSM and CLT to the study of information-sharing behaviours among social media users during crises, this paper offers valuable insights that can assist emergency management departments in effectively guiding online public opinion during crisis situations.

## 2. Literature Review and Theoretical Foundation

### 2.1. User Information-Sharing Behaviours in a Crisis

Information-sharing behaviours are a kind of social exchange behaviours, including the behaviour of posting information and forwarding others’ information [11]. Users’ information sharing in a crisis refers to the behaviour in which users share their personal experience or online crisis information with others through posting and forwarding. Almost 60% of people report that they regularly share information with others online [19]. Scholars have studied the influencing factors of information-sharing behaviours from the perspectives of information source [20], content [21], and receiver [22]. From the perspective of information content, scholars have mainly explored the impact of information representation and emotional attitudes on information-sharing behaviours in crises. For example, Zhang et al. [23] explored the role of images in information sharing and found that negative emotions from news can induce users to share their feelings. Apuke and Omar [24] studied disinformation sharing during COVID-19 and found that altruism was the most important predictor of information-sharing behaviours.

People in a crisis generate a lot of information content through social media, which inadvertently contains their complex psychological processes. These contents are expressed in different languages and quickly gathered on social media to stimulate the cognitive appraisal of social media users and the following emotional responses. Pennebaker et al. [25] found that people’s verbal descriptions inadvertently reflect their psychological world, and the external linguistic representation that reflects people’s psychological processes is called psychological language. More and more scholars have researched psychological language. Most of the data sources are self-reported, and some studies are based on social media such as X (formerly known as Twitter) and online forums. The topics cover psychological intervention [26], historical data analysis [27], personality traits [28], and language differences [29]. Compared with general situations, people’s decision making in a crisis is more likely to be affected by others on social media. However, the existing results have not paid enough attention to people’s psychology in the process of information sharing. Therefore, from the perspective of psychological language, this paper further studies the impact of different psychological processes on social media users’ information-sharing behaviours in a crisis. 

### 2.2. Heuristic–Systematic Model

During a crisis, the decision of whether to engage in information-sharing behaviours poses a complex decision making dilemma. In recent years, within the realm of decision making and reasoning research, the dual-system processing model, including notable frameworks such as the heuristic–systematic model (HSM) and the elaboration likelihood model (ELM), has garnered increasing scholarly interest. This theoretical system has established a comprehensive framework and behavioural paradigm for decision research, showcasing robust theoretical versatility and explanatory capabilities [30].

The HSM and the ELM are both part of an information processing model in social psychology, positing that individual information processing and attitude change can be categorised into two modes: heuristic and systematic. While the ELM places more emphasis on how individuals’ prior experiences and information processing motivations influence their responses, the HSM can be applied to a broader spectrum of information processing activities [31]. The HSM focuses on individuals’ immediate information processing capabilities and excels at adapting information processing modes based on the strength of individuals’ cognitive processing abilities in a given situation. Essentially, when individuals possess enough cognitive resources, they are more likely to engage in rational analytical processing for thorough information evaluation. In contrast, when cognitive resources are limited, individuals are inclined to rely on intuition-based heuristic thinking for quick decision making.

The HSM is a well-known communication model that seeks to elucidate how individuals receive and process persuasive messages. It has been applied to various contexts such as online shopping [32], tourism management [33], message communication [34], and consumer behaviour [35]. In recent years, the HSM has also found widespread application in the field of crisis management, offering valuable insights into individuals’ reactions to crises. For example, Rahmani and Kordrostami [36] employed the HSM to investigate changes in consumers’ online shopping behaviours during the pandemic, revealing that consumers may have relied on heuristics for quick purchasing decisions which were less sensitive to pricing. Zhu et al. [37] confirmed the impact of information processing on vehicle purchase intentions during crises, demonstrating that heuristic and systematic processing exert a significant and positive influence on purchase intentions. 

While initial research on the HSM suggested that analytical information processing is superior and that heuristic processing could yield biased outcomes, as opposed to rational analysis, recent studies have highlighted the ongoing debate regarding the benefits and drawbacks of heuristic versus analytical information processing [38]. Nevertheless, the current study has opted to employ the HSM framework because it offers a comprehensive perspective on how heuristic and systematic information processing influence information-sharing behaviours during crises, providing deeper insights into this aspect. 

### 2.3. Construal Level Theory 

CLT is a social psychology theory that explains the connection between psychological distance and an individual’s attitude and behaviour towards a forthcoming event [39], and it has been applied in contexts such as impulsive shopping [40], individual difference [41], public behaviour [42,43], crisis communication [39], etc. The theory put forward that individual cognitive differences can be conceptualised and operationalised as a psychological distance between the individual and the phenomenon [44].

Psychological distance refers to the “distance” factors that affect people’s different cognitive judgments and decisions [18], which usually takes the self as a reference point to interpret people’s reaction mechanisms to object cognition and evaluation decisions. It contains the important dimensions of time distance, space distance, and social distance. Psychological distance is influenced by the level of psychological interpretation and influences people’s predictions, preferences, and behaviours. People tend to have detailed and contextualised mental representations of low explanatory levels for things that are close to them and abstract and stable mental representations of high explanatory levels for things that are far away. With the extension of psychological distance, people are more inclined to conclude high-level construction. 

In a crisis, people are easily affected by the information content on social media and have the impulse to post or forward information, which leads to information-sharing behaviours. Due to the differences in time, place, and personal relevance, people may have different psychological distances for the same event. Different psychological distances will make people have different levels of interpretation for the same crisis and then have different impacts on their sharing behaviours. Public opinion communication is the information-sharing intention and behaviour generated after people perceive psychological distance [45]. In general, the closer the psychological distance, the more likely it is to arouse people’s attention [46] and the more likely to trigger the willingness to share information and, thus, the sharing behaviour.

Although CLT has been widely used to explain people’s perception judgment and behavioural decisions in various contexts, there still is criticism voiced around the theory. For example, Brügger [47] has highlighted concerns that researchers have, at times, extended the application of CLT beyond its intended scope. Given that CLT primarily focuses on transient mindsets and short-term processes, it may not be the most suitable tool for examining enduring changes. Nonetheless, this study has chosen to utilise CLT because it aids one in comprehending the distinct construal levels of individuals prompted by crises and facilitates an exploration of the differential impact of cognitive appraisal and emotional responses on information-sharing behaviours. 

## 3. Research Model and Hypotheses Development

### 3.1. Theoretical Model

During a crisis, individuals engage in cognitive appraisal based on their cognitive resources and past experiences [48]. Cognitive appraisal serves as the foundational framework of beliefs and perceptions through which individuals filter and interpret incoming information [49]. Particularly, in the context of crises, social media users are likely to generate primary and secondary cognitive appraisals [50]. Primary cognitive appraisal involves individuals’ assessment of the relationship between external stimuli and stress levels. This type of appraisal is typically influenced by heuristic information processing, as individuals may rely on simplified thinking due to constrained cognitive capacity during a crisis. On the other hand, secondary cognitive appraisal pertains to users’ evaluation of responses following reflection and contemplation after being triggered by a crisis. It entails a thorough and analytical examination of information relevant to judgment through systematic information processing. This mode of processing reflects an individual’s cognitive approach, depth of thinking, and cognitive processes, involving complex operational information.

Emotion, as a cognitive mode in humans, serves as the initial response to external stimuli, automatically triggered through the positive and negative emotional pools in the brain. This emotional reaction requires minimal cognitive effort and is part of the heuristic information processing process. Emotions can be described as psychological states that arise from the appraisal or evaluation of pertinent information [51]. The emotional response involves the process through which individuals associate emotions with various situations [52]. Positive and negative emotions have a significant impact on individuals’ perceptions and decision making processes. Positive emotions create feelings of happiness and delight, while negative emotions evoke sensations of fear, surprise, sadness, anger, and disgust, among others. These emotional responses play a crucial role in shaping individuals’ internal emotional judgments during a crisis.

When confronted with a challenging crisis, individuals typically engage in primary and secondary appraisal processes to assess the threat, challenge, or harm posed by the event, as well as the emergency management procedures in place. These appraisal processes lead to the generation of different emotional responses [53], which can subsequently influence information behaviours. In response to a crisis event, individuals often follow a cognition–emotion–behaviour sequence, where they first conduct a cognitive evaluation, followed by experiencing an emotional response, and finally form intentions and engage in behaviours [54]. This sequential process reflects how individuals process and respond to crisis stimuli, highlighting the interconnected nature of cognition, emotions, and behaviour in such situations.

According to the HSM, individuals engage in primary cognitive processing through heuristic cues when sharing what they observe and hear, and they utilise secondary cognitive processing through systematic cues when sharing their perspectives and opinions through rational thinking. Individuals who are not physically present at the scene of a crisis can only form mental images of the event based on the information shared by other social media users, including detailed descriptions and emotional tones. This information significantly influences people’s perceptions and subsequent behaviours. The information shared on social media encompasses both the initial cognitive processing and the deeper analysis of the crisis’s characteristics. These cognitive aspects contribute to users’ problem definition and causal explanations regarding the crisis event. Additionally, the emotional tone conveyed in the content influences users’ moral evaluations of the crisis-related information. Together, these cognitive and emotional dimensions in the information shared on social media shape users’ understanding and responses to crisis events.

During a crisis, individuals may exhibit positive or negative emotional responses based on heuristic cues following cognitive evaluation. These emotions can directly or indirectly impact their information-sharing behaviour. Emotions serve as adaptive responses to crisis events and typically stem from individuals’ subjective cognitive appraisal of the situation. Moreover, emotions play a pivotal role in driving individuals’ information-sharing behaviours, influencing the content they choose to share and how they communicate about the crisis. Emotions shape not only how individuals perceive and react to a crisis but also how they communicate and engage with others regarding the event. Therefore, the emotional response may play a mediating role between cognitive appraisal and information-sharing behaviour. As a mediating variable, the emotional response interweaves with cognitive appraisal, which jointly promote microblog users’ information-sharing behaviours.

In addition, people are in different times and places and pay different attention to events, which will produce different levels of time, space, or social distance from the events. CLT explains the connection between psychological distance and an individual’s attitude and behaviour towards a forthcoming event [39], which has been used to examine how psychological distance contributes to the public perception of a crisis. The central premise of CLT is that people’s mental construal of an event is shaped by psychological distance [55]. According to CLT, the closer the psychological distance and the lower the construal level, the stronger the direct experience of a crisis and the stronger the information-sharing behaviour. On the contrary, individuals with greater psychological distance and higher construal levels have a weaker direct experience, which weakens the influence of the cognitive appraisal and emotional response on information-sharing behaviours. 

In summary, the HSM provides a framework to predict how varying types of information processing can influence information-sharing behaviours. This model forms the basis for examining the impact of cognitive appraisal and emotional response on information-sharing behaviours. Additionally, CLT explains the moderating variable of the construal level and its role in influencing behaviour. The proposed theoretical model integrates these concepts and is depicted in Figure 1, illustrating the interplay between cognitive appraisal, emotional response, and information-sharing behaviours during a crisis event. 

### 3.2. Hypotheses’ Development

#### 3.2.1. The Influence of Cognitive Appraisal on Information-Sharing Behaviours

The HSM posits that human beings engage in two distinct cognitive thinking patterns: heuristic and systematic. Heuristic thinking is characterised as perceptual, intuitive, and unconscious, while systematic thinking is described as cognitive, controlled, and conscious. Heuristic processing involves quickly drawing conclusions based on a few informational cues and assessing the available information. In contrast, systematic processing requires a more significant cognitive effort and the conscious evaluation of information. According to the HSM, heuristic processing is the default mode that often prevails over systematic processing, as individuals tend to minimise cognitive efforts to conserve limited cognitive resources [16]. During a crisis, individuals exhibit varying degrees of cognitive appraisal of the emergency. Even when faced with the same external stimuli, people may engage in different psychological cognitive processes. This variability in cognitive appraisal can influence how individuals perceive, evaluate, and respond to a crisis event. The interplay between heuristic and systematic thinking plays a crucial role in shaping individuals’ cognitive processes and subsequent behaviours in crisis situations.

Heuristic information processing usually belongs to the primary stage of cognitive activities. It reflects the basic attributes of objective things produced by the information processing activities of the brain when objective stimuli act on the sensory organs. The information conveyed during emergencies reflects individuals’ primary cognitive appraisal, whereby their observations, auditory inputs, and feelings are articulated through a language which mirrors their initial cognitive processes. This primary cognitive response, often indicated by terms like noisy, harsh, dark, bitter, and dizzy, serves as a heuristic cue. This basic cognitive approach simplifies information into terms that can enhance information sharing. Additionally, systematic information processing involves a psychological process wherein individuals actively seek, receive, and organise information in a structured manner. In a crisis, people generate secondary cognitive appraisal processes through insight and introspection and express them through a language representing secondary cognitive psychological processes. As a systematic cue, secondary cognition is often characterised by terms like simply, believe, should, basis, root, and attribution, which can encourage users’ information-sharing behaviours. Therefore, I assume that, in a crisis, the following take place:

**H1a.** *The primary cognitive appraisal contained in the information content can positively affect users’ information-sharing behaviours*.

**H1b.** *The secondary cognitive appraisal contained in the information content can positively affect users’ information-sharing behaviours*.

#### 3.2.2. The Influence of Emotional Responses on Information-Sharing Behaviours

Emotional response is a psychological process that involves both cognitive and physiological reactions to external stimuli. It is characterised by a rapid onset, an involuntary nature, and a short duration and, therefore, falls under heuristic information processing. Emotions can be categorised into positive and negative emotions as separate and independent variables that do not influence one another. Emotions are subjective experiences that often have a significant impact on people’s attitudes and behaviours. When individuals evaluate potential threats or challenges using heuristic information processes, emotional responses are triggered and subsequently influence their behaviours [56]. 

Usually, because of the idea of value exchange, self-presentation, or reputation enhancement [57], people are willing to share information that is positive, optimistic, or makes others feel good. Therefore, users affected by the positive emotion will exhibit stronger information-sharing behaviours. In addition, the crisis will also induce people’s negative emotional response, further affecting decision making behaviours. For example, people’s “panic” and “fear” for emergencies, “pity” and “anxiety” for life, etc., lead to negative emotional experiences [58]. However, individuals who are infected by negative emotions tend to become more nervous. The stronger the negative emotional response, the easier it is for users to feel vigilant, which may inhibit information-sharing behaviours to a certain extent. As a heuristic cue, positive emotions are typically represented by words like warmth, approval, hero, encouragement, and beauty, reflecting the positive psychological responses of individuals. On the other hand, negative emotions are often conveyed by words such as worry, bad, protest, violence, irony, and anger to signify the negative psychological reactions of individuals. Therefore, I assume that, in a crisis, the following things happen:

**H2a.** *The positive emotional response contained in the information content can positively affect users’ information-sharing behaviours*.

**H2b.** *The negative emotional response contained in the information content can negatively affect users’ information-sharing behaviours*.

#### 3.2.3. The Moderating Effect of the Construal Level

CLT elucidates how distance is associated with the abstract mental construal of events [59]. According to the time interpretation theory of psychological distance [18], greater time intervals lead individuals to employ abstract features to portray events. Conversely, shorter periods are more likely to result in the representation of events with specific features [60]. In terms of spatial distance, compared with close-distance events, people are used to recalling spatially distant events in a more abstract language. The greater the spatial distance, the easier it is to produce a higher level of explanation for the events. Social distance describes the degree of intimacy or distance between people. Compared with events closer to oneself, distant events have a relatively low impact on individuals and make it easier to adopt a more abstract level of explanation. Different dimensions of psychological distance are related to one another in individual cognition, and these dimensions have similar effects on the level of psychological interpretation and affect people’s predictions, preferences, and actions.

In a crisis, individuals with greater psychological distance have a higher construal level of crisis events and are more able to generate mental representations of events through more abstract thinking modes and more thoroughly see the essence of events from a more comprehensive and broad perspective [46]. From a cognitive perspective, a high level of interpretation helps individuals pay more attention to the overall information, assess the severity of emergencies more objectively in crisis, and weaken the one-sided influence of cognitive appraisal on information-sharing behaviours. Therefore, I assume that, in a crisis, the following happen:

**H3a.** *The construal level weakens the influence of primary cognitive appraisal on information-sharing behaviours*.

**H3b.** *The construal level weakens the influence of secondary cognitive appraisal on information-sharing behaviours*.

From the perspective of emotion, individuals’ cognitive resources are limited in a crisis: individuals with a close psychological proximity have a lower construal level, and their behavioural judgments are more likely to be affected by positive or negative emotional responses; individuals with psychological distance have a higher construal level of crisis events, and a higher construal level of things is more likely to reduce the impact of the emotional response on the information-sharing behaviour. Therefore, I assume that, in a crisis, the following hold true:

**H4a.** *The construal level weakens the influence of a positive emotional response on the information-sharing behaviour*.

**H4b.** *The construal level weakens the influence of a negative emotional response on the information-sharing behaviour*.

#### 3.2.4. The Mediating Effect of Emotional Responses

A high uncertainty in information leads people to engage in two types of information processing during a crisis: heuristic and systematic. In a complex crisis, these processes may intertwine, presenting a more intricate influence on individuals’ information-sharing behaviours. The cognitive appraisal theory of emotions posits that emotions are contingent upon individuals’ appraisal of stressors, specifically how they are affected by primary and secondary appraisals [61]. During a crisis, individuals may experience varying objective feelings toward the events they are witnessing, hearing, and feeling. Following rapid heuristic information processing, a primary cognitive appraisal is formed, leading to rapid emotional processing. Posts with emotional responses are more likely to be followed by friends or other users and promote the sharing of information that the latter will post and repost. Primary cognitive appraisal may have an impact on users’ information-sharing behaviours either directly or through positive or negative emotional responses. Therefore, I assume that, in a crisis, the following happen:

**H5a.** *A positive emotional response plays a mediating role between primary cognitive appraisal and information-sharing behaviours*.

**H5b.** *A negative emotional response plays a mediating role between primary cognitive appraisal and information-sharing behaviours*.

When individuals encounter potential stressors, they utilise secondary cognitive appraisal to evaluate external risks using their systematic reasoning skills. This assessment is expressed through secondary cognitive appraisal, subsequently triggering emotional processing in individuals’ brains and eliciting the corresponding positive or negative emotions. The emotional information content influences individuals’ information-sharing behaviours. Specifically, secondary cognitive appraisal can impact users’ information-sharing behaviours either directly or through positive or negative emotional responses. Therefore, I assume that, in a crisis, the following take place:

**H6a.** *A positive emotional response plays a mediating role between secondary cognitive appraisal and information-sharing behaviours*.

**H6b.** *A negative emotional response plays a mediating role between secondary cognitive appraisal and information-sharing behaviours*.

## 4. Research Design

### 4.1. Sample Selection and Data Source

Tang et al. [62] pointed out that a single case is suitable for the in-depth analysis of a research object and for refining the law behind the replication phenomenon. The Jiuzhaigou earthquake in the Sichuan province occurred on the 8 August 2017. On the 13 August 2017, the number of casualties reached 550, with a total of 176,492 people affected and 73,671 houses damaged to varying degrees. This was a typical sudden disaster event, and Sina Weibo was a popular social media platform with wide representation. Therefore, this research took public opinion data on the “Jiuzhaigou Earthquake” on Sina Weibo as the research object, adopted the keyword-crawling function of aggregate search, and set “Jiuzhaigou earthquake” as the keyword for data search. 

According to the changing trends in microblog popularity, the data from the first post published on Sina Weibo after the earthquake to the gradual subsiding of the event were selected. A total of 60,890 Weibo public opinion data were obtained. By preprocessing the data and removing duplicate data and some irrelevant and abnormal data, the result was a total of 55,720 posts gathered. Text processing used TextMind to preprocess all the blog entries by word segmentation, removing stop words, outliers, etc., and then analysed the psycholinguistic dimensions in the text. Afterwards, the data were aggregated on a 1 h basis, and 353 h of data were obtained.

### 4.2. Variable Design

Psychological language is often used to explore the linguistic dimensions related to people’s perception, cognition, emotion, and other mental processes. This study analysed the text content generated by social media from the perspective of psychological language, which can better reveal the intrinsic nature of people’s information-sharing behaviours. LIWC (Linguistic Inquiry and Word Count) [63] is an effective tool for textual psychoanalysis through word metrics, using pre-validated dictionaries to capture themes and psychological states from texts. In the network context, LIWC has also been effectively verified.

Therefore, this study analysed the content through LIWC and obtained the words that reflected users’ primary cognition, secondary cognition, positive emotion, negative emotion, time, space, and personal concern, which were used as the measurement variable of the cognitive appraisal, emotional response, and construal level. Considering that the higher the construal level, the less relevant words will be used, the LIWC words corresponding to spatial distance, time distance, and social distance were reciprocal. For the information-sharing behaviour, the number of microblog posts and the number of retweets per unit time window were used as the measurement variables. In addition, to avoid missing variables causing estimation bias, the proportion of male users and the proportion of institutional users representing user characteristics were used as the control variables. In summary, the variables and their meanings are shown in Table 1.

## 5. Data Analysis

### 5.1. Descriptive Statistics and Correlation Analysis

The descriptive statistics of the variables involved are shown in Table 2. From the perspective of the descriptive statistics of the variables, the study samples differed greatly in the value of each variable. To eliminate the influence of dimensionality and magnitude between the variables, make variables more comparable, avoid unnecessary errors, ensure the reliability of results, and meet the conditions of normal distribution as much as possible, logarithmic processing of all the variables was carried out. 

### 5.2. Hypothesis Test

#### 5.2.1. Hypothesis Test Result 

As it can be seen from Table 2, some correlation coefficients between the variables were greater than 0.5, indicating that there may have been serious multicollinearity between the variables. Therefore, to avoid the problem of model estimation distortion caused by traditional least square estimation, the lasso regression method [64] was adopted. The method reduced the regression coefficient by punishing the size of the regression coefficient and selected the best parameter by ten-fold cross-validation. Spatial distance, time distance, and social distance were used as the moderating variables to test the effects of cognitive appraisal and emotional response on information-sharing behaviours. The results are shown in Table 3.

Furthermore, to test the mediating effects of positive and negative emotional responses, this study referred to the method of [65] and used the Sobel test to build a model. Among them, the mediating effects of positive and negative emotional responses on the primary and secondary cognitive appraisals and the information-sharing behaviours posted and forwarded by users are tested in Table 4 and Table 5.

#### 5.2.2. The Main Effects of Cognitive Appraisal and Emotional Responses on Information-Sharing Behaviours

As it can be seen from Table 3, both the primary cognitive and secondary cognitive appraisals had significant positive effects on the posting and forwarding behaviours, which supported hypotheses H1a and H1b. At the same time, the influence of secondary cognitive appraisal on the information-sharing behaviours was greater than that of primary cognitive appraisal. This is because cognitive appraisal is the cognitive structure of experience or knowledge that people store in the brain [50], and it is the characteristic description of objective things in people’s brains [66] through the double process of information processing to produce different cognitive degrees. In crises, people produce rapid primary cognitive responses to external stimuli and share the information they see, hear, and feel through social media to form a primary cognitive appraisal. However, with the progression of the event, more comprehensive external information becomes available, which enables us to have in-depth insight and thinking and further trace the cause and source of the event. With the deepening of information content on social media, the content of blog posts using secondary cognitive appraisal can better promote the information-sharing behaviour of users.

In addition, text content with positive emotions is more likely to be shared. The results in Table 3 further verify that a positive emotional response has a significant positive effect on the information-sharing behaviours of posting and forwarding, which supports hypothesis H2a. However, the influence coefficient of a negative emotional response on information-sharing behaviours is significantly positive, indicating that the adoption of a negative emotional response will also positively affect users’ information-sharing behaviour, so H2b has not been verified. At the same time, a positive emotional response has a greater impact on information-sharing behaviours than a negative emotional response. This is because people often post or retweet text content for self-presentation purposes, and most people prefer to share content that makes others feel good rather than sad or upset [67]. Therefore, compared with a negative emotional response, a positive emotional response can better promote users’ information-sharing behaviours. 

#### 5.2.3. The Moderating Effect Test of the Construal Level

It can also be seen from Table 3 that the interaction coefficients of spatial distance with secondary cognitive appraisal and a positive emotional response are all negative, indicating that spatial distance significantly negatively regulates the influence of secondary cognitive appraisal and a positive emotional response on information-sharing behaviours. So, H3b and H4a are verified. However, the moderating effect of time distance and social distance is not obvious. From the perspective of the interaction coefficient between spatial distance and primary cognitive appraisal, only the coefficient of the influence on the posting behaviour is significantly negative, but relatively small (−0.010), and the moderating effect on the forwarding behaviour is not significant, so H3a has been partially verified. In addition, from the perspective of the interaction coefficient between spatial distance and a negative emotional response, the influence coefficient on the forwarding behaviour is significantly negative, and the moderating effect on the posting behaviour is not obvious, so H4b has also been partially verified.

#### 5.2.4. The Mediating Effect Test of the Emotional Response

Table 4 shows the mediating role of a positive emotional response between primary cognitive appraisal and information-sharing behaviours. Models 16 and 18 show that primary cognitive appraisal has a significant positive impact on users’ posting and forwarding behaviours, and model 15 shows that primary cognitive appraisal has a significant positive impact on a positive emotional response. After adding a positive emotional response to models 17 and 19, primary cognitive appraisal still has a significant positive effect on the posting and forwarding behaviours, but the regression coefficient is significantly reduced, indicating that a positive emotional response plays a partial mediating role in the relationship between primary cognitive appraisal and information-sharing behaviours.

It can be seen from the results in Table 4 that the regression coefficient between secondary cognitive appraisal and information-sharing behaviours also decreases after the addition of a positive emotional response, indicating that a positive emotional response partially mediates the relationship between secondary cognitive appraisal and information-sharing behaviours. Similarly, the results in Table 5 show that a negative emotional response also has a partial mediating effect between primary and secondary cognitive appraisal and information-sharing behaviours. As a result, H5a, H5b, H6a, and H6b are all partially supported.

## 6. Discussion

Integrated with the HSM and CLT, this paper examines the influence of cognition and emotion on users’ information-sharing behaviours during a crisis. Additionally, this study investigates the moderating role of the construal level and the mediating role of the emotional response. The findings enhance the utilisation of the HSM and CLT in crisis information management and offer important insights for preventing and addressing public opinion risks on social media during crises. The results are presented in Table 6.

### 6.1. Discussion

As it can be seen from Table 6, H2b is not supported. This may be because an emotional response is a person’s inner moral evaluation [68]. Under crises, people have different emotional experiences, and the content of blog posts covers different emotional tones. A positive energy in people’s hearts leads to the creation of blog posts with a positive response to promote information sharing. However, an inner negative emotion leads the blogger to a negative response and also triggers the information-sharing behaviour of the public. In addition, emotional information content in a crisis is more likely to cause physiological arousal, which has been proven to be the intrinsic driving force behind information-sharing behaviours [67]. Compared with neutral or positive emotions, people with negative emotions are more inclined to fix their bad emotions. In the context of social media, posting and forwarding blog posts can effectively resolve or channel negative emotions. Therefore, adopting a negative emotional response will also positively promote users’ information-sharing behaviours.

Here, H3a and H4b are partially supported. On the one hand, this may be because posting is an original behaviour. In a crisis, people are more willing to share descriptive original blog information for the first time. However, with the increase in spatial psychological distance, people have a higher construal level and adopt a more abstract level to explain primary cognitive appraisal, which weakens the impact on the posting behaviour. On the other hand, after being affected by emotional responses, posts with emotions are more likely to be forwarded [69]. 

Similarly, with the extension of spatial psychological distance, people’s abstract ability is stronger, and they are more able to make high-level psychological constructions, weakening the influence on the forwarding behaviour. Therefore, construal level distance weakens the influence of primary cognitive appraisal on posting behaviours and the influence of a negative emotional response on reposting behaviours. However, the moderating effect of time distance and social distance is not obvious. Although the spatial, temporal, and social dimensions of psychological distance have the same influence on the level of psychological interpretation, the spatial distance and the other two dimensions have a certain asymmetry [70]. Compared with other psychological distance dimensions, spatial distance is the basic and main psychological dimension, which leads to people’s greater distance perception and affects people’s psychological perception of the time distance and social distance to a certain extent. Therefore, the regulating effect of spatial distance becomes more obvious. 

In addition, the mediating role of the emotional response is also partially supported. Previous studies have found that, when people see information related to emergencies posted by different information sources on social media, they are more likely to produce primary and secondary cognitive responses. People produce cognitive responses through the information processing process of the brain. Gradually, primary and secondary cognitive appraisals are formed on social media [50], which may also have a psychological impact on users and be expressed through positive or negative emotional language, and then gradually gather to form an emotional response [52] and have an impact on information-sharing behaviours. People’s cognitive and emotional responses may become intertwined and work together to promote users’ information-sharing behaviours. Therefore, positive and negative emotional responses play a partially mediating role between primary and secondary cognitive appraisal and information-sharing behaviours.

In the research results of this paper, there are two new findings.

(1) It has been found that cognitive appraisal and emotional response have a significant positive impact on information-sharing behaviours in a crisis. Secondary cognitive appraisal consumes more cognitive resources, pays more attention to describing the potential meaning of crises, analyses emergencies more thoroughly, and has a stronger impact on users’ information-sharing behaviours. In a crisis, people are more willing to share positive information and avoid content with negative information [67]. Compared with negative emotions, positive emotions are more likely to cause viral transmission. Therefore, in an emotional response, the adoption of positive emotions has a stronger impact on information-sharing behaviours.

(2) It has been found that the construal level of users in crisis has a moderating effect between the information content and the sharing behaviour, and the emotional response has a partial mediating effect between cognitive appraisal and the information-sharing behaviour. The three measurement dimensions of the construal level, namely, spatial distance, time distance, and social distance, have different moderating effects. In a crisis, the spatial distance perceived by users is the most basic dimension of the construal level [70] and has the most obvious moderating effect. People with a long spatial distance tend to adopt a more abstract and stable high level of interpretation of information related to the crisis, which restrains the influence of cognitive appraisal and the emotional response on information-sharing behaviours to a certain extent. As a mediating variable, the emotional response interweaves with the cognitive appraisal, which jointly promote users’ information-sharing behaviours. 

### 6.2. Theoretical Contribution

By integrating the HSM and CLT, this study presents a comprehensive framework that explores the impact of cognition and emotions on users’ information-sharing behaviours during a crisis. This research identifies two emotional outcomes resulting from primary and secondary cognitive appraisal—positive and negative emotional responses—which subsequently influence information-sharing behaviours. This study also considers the differentiated effects of the different construal levels of the individuals in these processes.

Three main theoretical contributions emerge from this research. Firstly, the extension of the HSM to investigate users’ information-sharing behaviours in crises provides valuable insights. This study examines the influence of cognition and emotions as heuristic or systematic cues on social media users’ information-sharing behaviours during a crisis, offering a model of information-sharing behaviours. The findings reveal that both heuristic and systematic cues have significant positive effects on information-sharing behaviours during a crisis. The extension of the HSM to emergency events’ crisis scenarios further expands its application beyond the traditional realm of risk decision making, demonstrating its relevance in the context of social media during crises.

Secondly, this study uncovers the moderating role of the construal level between cognitive appraisal, emotional response, and information-sharing behaviours during crises. This paper studies the influence of various dimensions of CLT [18] on information content and information-sharing behaviours in a crisis and verifies the fundamental role of spatial distance in psychological distance and on people’s psychological construction level [70]. It enriches the research on the information-sharing behaviour of individual-differentiated influences and extends the CLT from the perspective of information content in a crisis.

Thirdly, this study subdivides cognition and emotion and uses them as heuristic or systematic cues to explore their differentiated effects on information-sharing behaviours. The findings indicate that the impact of systematic cues on information-sharing behaviours is significantly more pronounced than that of heuristic cues during a crisis. Among the heuristic cues, positive emotions exert a greater influence on information-sharing behaviours compared to primary cognition and negative emotions. This study further verified the mediating role of positive and negative emotional responses between fine-granularity cognitive appraisal and information-sharing behaviours and clarified the relationship between users’ cognitive appraisal, emotional response, and information-sharing behaviours on social media platforms.

### 6.3. Practical Implications

This study has some implications for the management practice of the government and emergency management departments. On the one hand, it is necessary to fully consider the influence of the information content on social media and pay attention to secondary cognitive appraisal and positive emotional responses for crisis communication with the public, guide and control online public opinion, and pay timely attention to primary cognitive appraisal on social media. This allows one to effectively analyse the emotional tendencies contained in the content of a blog and effectively find out the root cause of the public’s negative emotions towards the progress of an emergency in a crisis. 

On the other hand, it is necessary to fully monitor the construal level inadvertently shown by the public through blog content in crises, especially paying attention to the blog content generated by users with a close psychological distance, identify the negative emotional tendencies expressed by these users through their text content as early as possible, effectively guide them to form benign cognition and positive emotions, and, through a more positive energy in the network, stimulate people to produce a higher level of psychological construction and more effectively inhibit the irrational spread of negative emotions.

### 6.4. Limitations

This research also has certain limitations. First, it is essential to broaden the scope beyond purely examining the effects of cognition and emotion on information content. Future studies should also encompass a comprehensive analysis of how information content, information sources, and contextual factors collectively influence information-sharing behaviours. Second, investigating the nuanced impact of specific emotions such as happiness, joy, anger, sadness, fear, disgust, and surprise on users’ information-sharing behaviours holds great research relevance and importance. Third, further attention should be paid to text content covering some functional words such as pronouns, adverbs, interjections, and terms such as severity, sensitivity, and uncertainty to carry out the in-depth mining of the influence path of information content on information-sharing behaviours. Furthermore, it is imperative to conduct further thorough research considering the stage of a crisis, as heuristic and systematic information processing may exert varying influences on the information-sharing behaviours during different crisis stages.

## 7. Conclusions

This study utilised the HSM in conjunction with CLT to incorporate considerations of individuals’ construal levels into the research model of information-sharing behaviours during crises. By merging these theoretical frameworks, this study further divided cognitive appraisal and emotional responses during crises and put forth a theoretical model elucidating social media users’ information-sharing behaviours. Additionally, it delved into the moderating role of the construal level and the mediating effect of emotional responses on these behaviours. The results show that both cognition and emotions promote users’ information-sharing behaviour through heuristic or systematic information processing. Secondary cognition, as a systematic cue, has a greater impact on information-sharing behaviours, and the impact of positive emotions on heuristic cues is greater than that of primary cognition and negative emotions. However, a higher level of interpretation may weaken the influence of cognition and emotions on information-sharing behaviours. This research expanded the application of the HSM and CLT to crisis contexts. Moreover, future studies should explore additional variables and potential factors that could impact information-sharing behaviours during crises.

## Figures and Tables

**Figure 1 behavsci-14-00495-f001:**
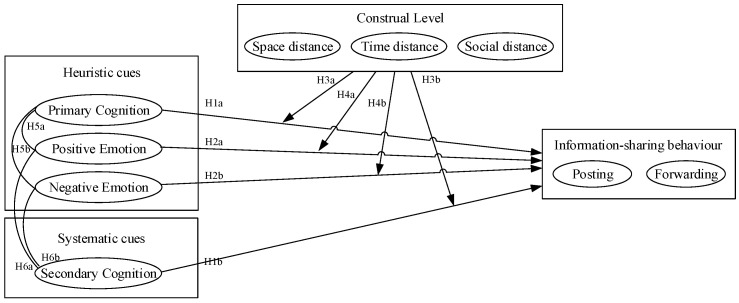
Research model.

**Table 1 behavsci-14-00495-t001:** Variables’ design.

Variable	Measurement	LIWC Words	References
Primary cognitive appraisal (PCA)	Words that reflect dimensions such as those which are sensory, perceptual, etc.	Percept	[50]
Secondary cognitive appraisal (SCA)	Words that reflect dimensions such as insight, cause, etc.	CogMech	
Positive emotional response (PER)	Words that reflect positive emotions such as happiness, blessing, etc.	PosEmo	[52]
Negative emotional response (NER)	Word that reflects negative emotions such as sadness, anger, etc.	NegEmo	
Space distance (SP_PD)	Words that reflect the location of the event.	Space	[46]
Time distance (TM_PD)	Words that reflect the time of the event.	Time	
Social distance (SC_PD)	Words that reflect personal concerns.	I	
Posting behaviour (PST)	The number of Weibo posts.	/	[1]
Forwarding behaviour (FWD)	The number of Weibo retweets.	/	
Ratio of male to female users (RMF)	Control variable: ratio of male to female users.		
Ratio of institutional to individual users (RII)	Control variable: proportion of institutional individual users.		

**Table 2 behavsci-14-00495-t002:** Descriptive statistics.

Variables	PCA	SCA	PER	NER	SP_PD	TM_PD	SC_PD	RMF	RII	PST	FWD
PCA	1										
SCA	0.973	1									
PER	0.945	0.957	1								
NER	0.938	0.949	0.942	1							
SP_PD	−0.870	−0.904	−0.857	−0.833	1						
TM_PD	−0.796	−0.840	−0.770	−0.778	0.922	1					
SC_PD	−0.336	−0.374	−0.305	−0.313	0.480	0.458	1				
RMF	−0.014	−0.016	−0.024	0.008	−0.025	0.023	0.221	1			
RII	0.467	0.482	0.438	0.462	−0.485	−0.449	−0.212	0.229	1		
PST	0.969	0.9856	0.972	0.955	−0.889	−0.807	−0.334	0.009	0.504	1	
FWD	0.832	0.846	0.827	0.823	−0.756	−0.679	−0.309	0.042	0.500	0.859	1
Min	0	0.693	0	0	0.110	0.446	0.169	0	0	0.693	0
Max	6.909	9.240	8.212	6.759	0.621	1.899	100,000,000	100	75	7.742	9.835
Mean	3.747	5.674	4.314	3.107	0.211	0.606	4,815,864	55.125	17.981	3.972	4.089
SD	1.456	1.504	1.526	1.653	0.075	0.136	21,400,000	13.122	14.395	1.512	2.680

**Table 3 behavsci-14-00495-t003:** The result of the hypothesis test.

Independent Variable	Dependent Variable: Information-Sharing Behaviour													
	PST	FWD	PST		FWD		PST		FWD		PST		FWD	
	Model 1	Model 2	Model 3	Model 4	Model 5	Model 6	Model 7	Model 8	Model 9	Model 10	Model 11	Model 12	Model 13	Model 14
PCA	0.074	0.182	0.085	ns	0.178	0.033	0.073	0.048	0.084	0.091	0.081	0.07	0.181	0.057
SCA	0.527	0.734	0.516	0.625	0.736	0.856	0.541	0.668	1.035	0.997	0.533	0.555	0.723	0.941
PER	0.293	0.312	0.314	0.389	0.306	0.382	0.315	0.29	0.298	0.303	0.307	0.299	0.309	0.29
NER	0.061	0.189	0.065	0.025	0.189	0.284	0.058	0.043	0.143	0.145	0.063	0.06	0.19	0.138
SP_PD			−0.026	1.026	ns	1.467								
PCA*SP_PD				−0.01		ns								
SCA*SP_PD				−0.89		−0.854								
PER*SP_PD				−0.878		−1.152								
NER*SP_PD				ns		−0.672								
TM_PD							0.108	0.283	1.89	1.596				
PCA*TM_PD								ns		ns				
SCA*TM_PD								−0.205		ns				
PER*TM_PD								ns		ns				
NER*TM_PD								ns		ns				
SC_PD											ns	ns	ns	ns
PCA*SC_PD												ns		ns
SCA*SC_PD												ns		ns
PER*SC_PD												ns		ns
NER*SC_PD												ns		ns
RMF	ns	0.005	0.002	0.001	0.004	0.004	0.001	0.001	0.005	0.005	0.001	0.001	0.004	0.003
RII	0.002	0.022	0.004	0.003	0.021	0.019	0.004	0.004	0.023	0.022	0.004	0.004	0.021	0.022
_cons	−0.791	−3.364	−0.992	0.066	−3.27	−2.032	−1.136	−1.017	−5.68	−5.312	−1.026	−1.044	−3.211	−3.67
Lambda	36.881	2.604	0.893	0.893	12.662	6.602	3.579	3.928	0.335	4.537	1.712	2.726	16.739	1.635
MSPE	0.042	1.99	0.039	0.028	1.99	1.975	0.04	0.039	1.959	1.975	0.038	0.038	1.992	1.991
St. dev.	0.005	0.194	0.005	0.003	0.193	0.189	0.004	0.005	0.292	0.296	0.005	0.005	0.193	0.196

Note: ns means not significant.

**Table 4 behavsci-14-00495-t004:** The mediating effect of positive emotional responses between cognitive appraisal and information-sharing behaviour.

Independent Variable	PER		Dependent Variable: Information-Sharing Behaviour							
			PST		FWD		PST		FWD	
	Model 15	Model 20	Model 16	Model 17	Model 18	Model 19	Model 21	Model 22	Model 23	Model 24
PCA	0.9608		0.9662	0.4391	1.2468	0.7447				
SCA		0.9448					0.9348	0.6223	1.2457	0.9895
PER	/	/	/	0.5065	/	0.6768	/	0.3288	/	0.26
RMF	ns	ns	ns	ns	ns	0.0052	ns	ns	ns	ns
RII	ns	ns	0.0062	0.0039	0.0099	0.0252	ns	0.0023	0.0057	0.0066
_cons	0.7133	−1.0473	0.2406	0.0716	−0.7614	−2.3605	−1.3318	−1.019	−3.0815	−2.7643
Lambda	29.621	27.343	10.813	48.096	244.529	0.437	58.725	40.477	248.567	248.567
MSPE	0.2547	0.1987	0.136	0.0715	2.3015	2.032	0.0733	0.0444	2.1631	2.1459
St. dev.	0.0198	0.024	0.0117	0.0072	0.1557	0.1818	0.0073	0.0048	0.163	0.159

Note: ns means not significant.

**Table 5 behavsci-14-00495-t005:** The mediating effect of negative emotional responses between cognitive appraisal and information-sharing behaviour.

Independent Variable	NER		Dependent Variable: Information-Sharing Behaviour							
			PST		FWD		PST		FWD	
	Model 25	Model 30	Model 26	Model 27	Model 28	Model 29	Model 31	Model 32	Model 33	Model 34
PCA	1.0333		0.9662	0.6113	1.2468	0.7672				
SCA		1.0383					0.9348	0.7906	1.2457	1.0211
NER	/	/	/	0.3423	/	0.4426	/	0.1723	/	0.2159
RMF	ns	0.0024	ns	ns	ns	ns	ns	0.0009	ns	ns
RII	0.0021	ns	0.0062	0.0053	0.0099	0.0068	ns	0.0032	0.0057	0.0056
_cons	−0.8021	−2.9147	0.2406	0.5219	−0.7614	−0.2824	−1.3318	−1.1546	−3.0815	−2.4756
Lambda	21.954	5.506	10.813	5.137	244.529	268.371	58.725	5.737	248.567	248.567
MSPE	0.3356	0.2751	0.136	0.0979	2.3015	2.2624	0.0733	0.0566	2.1631	2.1602
St. dev.	0.0325	0.0321	0.0117	0.0096	0.1557	0.167	0.0073	0.0077	0.163	0.1682

Note: ns means not significant.

**Table 6 behavsci-14-00495-t006:** Results of hypothesis testing.

Hypothesis		Results
H1a	The primary cognitive appraisal contained in the information content can positively affect users’ information-sharing behaviours.	Support
H1b	The secondary cognitive appraisal contained in the information content can positively affect users’ information-sharing behaviours.	Support
H2a	The positive emotional response contained in the information content can positively affect users’ information-sharing behaviours.	Support
H2b	The negative emotional response contained in the information content can negatively affect users’ information-sharing behaviours.	No support
H3a	The construal level weakens the influence of primary cognitive appraisal on information-sharing behaviours.	Partial support
H3b	The construal level weakens the influence of secondary cognitive appraisal on information-sharing behaviours.	Support
H4a	The construal level weakens the influence of a positive emotional response on information-sharing behaviours.	Support
H4b	The construal level weakens the influence of a negative emotional response on information-sharing behaviours.	Partial support
H5a	A positive emotional response plays a mediating role between primary cognitive appraisal and information-sharing behaviours.	Partial support
H5b	A negative emotional response plays a mediating role between primary cognitive appraisal and information-sharing behaviours.	Partial support
H6a	A positive emotional response plays a mediating role between secondary cognitive appraisal and information-sharing behaviours.	Partial support
H6b	A negative emotional response plays a mediating role between secondary cognitive appraisal and information-sharing behaviours.	Partial support

## Data Availability

The data are available from the authors; however, they are not publicly available. Interested researchers may contact the corresponding author for access to the datasets.

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
