# Peer review of "The Influence of Cognitive and Emotional Factors on Social Media Users’ Information-Sharing Behaviours during Crises: The Moderating Role of the Construal Level and the Mediating Role of the Emotional Response"

_behavsci, 2024, doi:10.3390/bs14060495_

Round 1

Reviewer 1 Report

Comments and Suggestions for Authors

See attached file for improvement suggestions.

Comments on the Quality of English Language

No issues detected.

Minor editing necessary.

Author Response

R1: Insert one space between words and bibliographical data in brackets throughout the study.

Authors’ Response:

Thank you for your constructive feedback and encouragement. To clarify and enhance my research, the format check was carried out throughout the study, and one space was inserted between words and bibliographical data in brackets.

R2: Abstract: Please change one of them; avoid repetitiveness.

Authors’ Response:

Thank you for your constructive feedback and encouragement. To clarify and enhance my research, the abstract has been updated, and the word ‘intricate’ has been modified into ‘sophisticated’ to avoid repetitiveness.

R3: Research Model and Hypotheses Development: I think this is the third or fourth time you are repeating this statement. Which blog? This sentence needs reformulation: it has too many main sentences. These two sentences seem coming out of nowhere. They do not fit in the text.

Authors’ Response:

Thank you for your constructive feedback and encouragement. To clarify and enhance my research, the repeated statement in the first paragraph of Section 3.1 has been deleted. In addition, blog here is a general term for posts, not a specific post. To avoid ambiguity, it was modified to ‘The information content on social media’. Furthermore, the last sentence of the Section 3.2.1 has been reformulated. And I deleted and modified the unfit sentence in Section 3.2.2.

R4: Research Design: This sentence is incomplete. I am not sure this makes sense. If it does, it needs more additional explanation.

Authors’ Response:

Thank you for your constructive feedback and encouragement. To clarify and enhance the impact of my research, the sentence in Section 4.1 was replenished. Furthermore, the data were summarized according to the unit of 1 hour, so that the changing trend of information-sharing behaviour could be better observed. For ease of understanding, the description has been simplified in the last sentence of Section 4.1.

Thank you for your constructive advice again.

Reviewer 2 Report

Comments and Suggestions for Authors

The theoretical basis for building hypotheses is weak. Theories in the field must first be discussed, the knowledge gap identified, and then the study model must be built.

While the paper delves into the moderating role of construal level and the mediating role of emotional response, it may not cover other potential factors or theories that could also influence information-sharing behaviors during crises.

The paper relies on the cognitive appraisal theory of emotion (CATE) and construal level theory (CLT). While these theories are well-established, there might be criticisms or alternative perspectives within these frameworks that are not addressed. A more comprehensive discussion or comparison with other theories could strengthen the theoretical underpinnings.

Additional comments:

Main question addressed by the research: What is the factors that influence Social Media Users' Information-Sharing Behaviors during Crises?

The contribution is not clear. The theoretical basis for building hypotheses is weak. Theories in the field must first be discussed, the knowledge gap identified, and then the study model must be built.

While the paper delves into the moderating role of construal level and the mediating role of emotional response, it may not cover other potential factors or theories that could also influence information-sharing behaviors during crises.

It adds nothing new to the subject area compared with other published material, the paper relies on the cognitive appraisal theory of emotion (CATE) and construal level theory (CLT). While these theories have been discussed in many papers, the authors have to use new theories.

The problem not about the methodology, the problem about the study model, it is weak model. It is crucial to include a dedicated section in the literature review that explains the foundational theories supporting the study. This section should delve into the theoretical frameworks that form the basis of the research, offering essential context for developing the subsequent model. the authors must to review more theories in the field to add new variables.

The conclusion is missing.

The references are ok.

The quality of Figure One is poor.

Author Response

R1: The theoretical basis for building hypotheses is weak. Theories in the field must first be discussed, the knowledge gap identified, and then the study model must be built.

Authors’ Response:

Thank you for your constructive feedback and encouragement. To enhance the theoretical basis proposed by the research hypothesis, I have modified the theoretical basis of Section 2 and the research model and hypothesis of Section 3, specifically modified as follows:

Firstly, in Sections 2.2 and 2.3, CATE and CLT theories are systematically introduced to expand the application of the theories in different fields and scholars' critical opinions on the theories. Then, in Section 3.1, CATE and CLT theories are used to discuss the impact of cognition and emotion on information-sharing behaviour and the differentiated impact of different construal levels in combination with crises. Then, the specific role of the theories in constructing the research model of information-sharing behaviour is expounded. Finally, in Section 3.2, CATE and CLT theories are further integrated into the research problem, and the research hypothesis of response is proposed in combination with specific situations.

Thank you for your constructive advice again.

R2: While the paper delves into the moderating role of construal level and the mediating role of emotional response, it may not cover other potential factors or theories that could also influence information-sharing behaviours during crises.

Authors’ Response:

Thank you for your constructive feedback and encouragement. To clarify and enhance my research, some potential factors or theories that could also influence information-sharing behaviours during crises have been added to the second paragraph of the Introduction, which can be seen as follows:

Scholars have conducted in-depth research on information-sharing behaviour. Previous research developed the research model by involving the variables that are derived from different theoretical perspectives such as the theory of reasoned action (TRA) [2], theory of planned behaviour (TPB) [3], social cognitive theory [4, 5], uses and gratifications theory [6], risk management theory and protection motivation theory [7], self-determination theory [8], elaboration likelihood model (ELM) [9], etc. The fundamental basis of the theories usually pivots around two types of decision-making. One is the reasoned action model, which states that behaviour stems from a rational decision-making process, another is heuristic, implying that behaviour may ensue from emotional reactions to certain situations [10]. Attitudes and subjective norms usually are antecedents of behaviours in the reasoned path. While psychological images are usually in the social reactive pathway to explain the behaviour.

Thank you for your constructive advice again.

R3: The paper relies on the cognitive appraisal theory of emotion (CATE) and construal level theory (CLT). While these theories are well-established, there might be criticisms or alternative perspectives within these frameworks that are not addressed. A more comprehensive discussion or comparison with other theories could strengthen the theoretical underpinnings.

Authors’ Response:

Thank you for your constructive feedback and encouragement. To clarify and enhance my research, I revised Section 2.2 and Section 2.3 to strengthen the theoretical basis. Furthermore, the systematic evaluation of theories including criticisms has been modified.

A more comprehensive discussion about CATE has been added to the last paragraph of Section 2.2, which can be seen as follows:

By applying the CATE to the study of information-sharing behaviour in crises, a deeper understanding of individuals’ cognitive appraisals and emotional reactions can be attained. Although the CATE focuses on not only how emotions are generated but also explains its influence role on individual behaviour [39], there are still some criticisms of CATE, with limitations being highlighted. For example, Manthiou et al. [40] point out that emotional responses and consumer behaviour perform in the brain in a message input-output way, CATE illustrates only subjective judgments and emotional responses and ignores the mnemonic values that emotions can bring about. However, the current study has selected to use the CATE because it helps to understand the underlying motivational and evaluative roots of emotions and can predict how elicited emotions may influence behaviour [17].

A more comprehensive discussion about CLT has been added to the last paragraph of Section 2.3, which can be seen as follows:

Although CLT has been widely used to explain people's perceptions judgment and behavioural decisions in various contexts, there still is criticism voice about the theory. For example, Brügger [49] points out that researchers have sometimes overextended the application of CLT, which mainly focuses on transient mindsets and short-lived processes, it is not the right tool to examine long-lasting changes. However, this study has still selected to use the CLT because it helps to understand the different construal levels of people triggered by crises, and then the differentiated impact of cognitive appraisal and emotional response on information-sharing behaviour.

R4: The contribution is not clear. It adds nothing new to the subject area compared with other published material, the paper relies on the cognitive appraisal theory of emotion (CATE) and construal level theory (CLT). While these theories have been discussed in many papers, the authors have to use new theories or add new variables.

Authors’ Response:

Thank you for your constructive feedback and encouragement. To clarify the contribution, I reexamined the literature on using the CATE and CLT in the context of crises. Based on the result, I modified the last paragraph of the Introduction and highlighted the contribution of my study. Furthermore, the contribution has been reexamined and modified in Section 6.2 and Conclusion.

However, I am so sorry that I did not add new variables or use new theories to the study. The reasons are as follows:

First, CATE can help to understand the underlying motivational and evaluative roots of emotions and can predict how elicited emotions may influence behaviour, and CLT helps to understand the different construal levels of people triggered by crises, and then the differentiated impact of cognitive appraisal and emotional response on information-sharing behaviour. Second, to the best of my knowledge, this study is the first to integrate CATE and CLT empirically into the context of information-sharing behaviour during a crisis, thereby creating opportunities for future research in information behaviour. Third, it’s also too difficult to contain more variables in the current model. To better illustrate this question, I added the research limitation to the section of the conclusion.

Thank you for your constructive advice again.

R5: The conclusion is missing.

Authors’ Response:

Thank you for your constructive feedback and encouragement. To clarify and enhance my research, the conclusion has been supplemented, which can be seen as follows:

  1. Conclusion

This study uses the CATE as a lens to integrate considerations of the individuals’ construal level into the research model of information-sharing behaviour. By integrating the CATE and CLT, this study subdivided cognitive appraisal and emotional response in crisis and proposed a theoretical model of social media users' information-sharing behaviours. Furthermore, it examines the moderating role of the construal level and the mediating effect of emotional response. The results show that both cognitive appraisal and emotional response have a promoting role on users' information-sharing behaviours, while a higher construal level would dilute the effect. The results extend the CATE and CLT to the context of crisis. Adding new variables and other potential factors that could also influence information-sharing behaviours during crises is essential in future research.

R6: The quality of Figure One is poor.

Authors’ Response:

Thank you for your constructive feedback and encouragement. To clarify and enhance my research, detailed information has been added to Figure 1. The modified figure is as follows.

Thank you for your constructive advice again.

Round 2

Reviewer 2 Report

Comments and Suggestions for Authors

The theoretical basis for building hypotheses is weak. Theories in the field must first be discussed, and then the study model must be built.

While the paper delves into the moderating role of construal level and the mediating role of emotional response, it may not cover other potential factors or theories that could also influence information-sharing behaviors during crises. 

The paper relies on the cognitive appraisal theory of emotion (CATE) and construal level theory (CLT). While these theories are well-established, there might be criticisms or alternative perspectives within these frameworks that are not addressed. A more comprehensive discussion or comparison with other theories could strengthen the theoretical underpinnings.

1. What is the main question addressed by the research?

What is the factors  that influence Social Media Users' Information-Sharing Behaviors during Crises?

2. Do you consider the topic original or relevant in the field? Does it address a specific gap in the field?

No, the contribution is not clear. The theoretical basis for building hypotheses is weak. Theories in the field must first be discussed, the knowledge gap identified, and then the study model must be built.

While the paper delves into the moderating role of construal level and the mediating role of emotional response, it may not cover other potential factors or theories that could also influence information-sharing behaviors during crises. 

3. What does it add to the subject area compared with other published material?

nothing new, the paper relies on the cognitive appraisal theory of emotion (CATE) and construal level theory (CLT). While these theories have been discussed in many papers, the authors have to use new theories.

4. What specific improvements should the authors consider regarding the

methodology? What further controls should be considered?

The problem not about the methodology, the problem about the study model, it is weak model. It is crucial to include a dedicated section in the literature review that explains the foundational theories supporting the study. This section should delve into the theoretical frameworks that form the basis of the research, offering essential context for developing the subsequent model. the authors must to review more theories in the field to add new variables.

5. Are the conclusions consistent with the evidence and arguments presented

and do they address the main question posed?

No, the conclusion is missing

6. Are the references appropriate?

The references are ok

7. Please include any additional comments on the tables and figures.

The quality of Figure One is poor.

Author Response

Response to Reviewer’s Comments

Dear reviewer, I want to express my sincere gratitude for your thorough evaluation of my research paper and the valuable feedback you provided. Your insights have been incredibly helpful in refining the study and addressing the key aspects that needed improvement. Below, I provide a more detailed response to each of the points you raised.

R1: What is the factors that influence Social Media Users' Information-Sharing Behaviors during Crises?

Authors’ Response: Thank you for your constructive feedback and encouragement. In response to your suggestion to clarify the multiplicity of factors influencing information-sharing behaviors, I have expanded the explanation in the revised version. As you pointed out, scholars have examined various perspectives, including information sources, content, and recipients to understand these behaviors.

In my study, I focus on analyzing the cognitive and emotional aspects embedded in information content. To address this more explicitly, I have modified the paper's title and supplemented the literature review section with a discussion on the diverse factors influencing information-sharing behaviors.

Thank you for bringing this to my attention, and I hope these modifications effectively convey the comprehensive analysis of factors impacting information sharing in crises.

R2: The contribution is not clear. The theoretical basis for building hypotheses is weak. Theories in the field must first be discussed, the knowledge gap identified, and then the study model must be built. While the paper delves into the moderating role of construal level and the mediating role of emotional response, it may not cover other potential factors or theories that could also influence information-sharing behaviors during crises.  

Authors’ Response: Thank you for your insightful feedback on my research paper. In response to your comments, I have conducted further literature analysis to better highlight the contributions of the study. Integrating my research questions, I have delved deeper into the literature to elucidate the significance of the Heuristic-Systematic Model (HSM) in this study. The theory's perspective on psychological processes offers a valuable framework for unlocking the "black box" of social media users' information-sharing behaviors during crises, as discussed in the introduction.

Building on this foundation, I have made significant revisions to the theoretical framework in the Abstract, the second section, the research model, and the hypotheses formulation in the third section. Through a more in-depth analysis, I have refined and modified the study's contributions to enhance clarity and relevance. While focusing on cognitive and emotional factors, I acknowledge the importance of considering other variables influencing information-sharing behaviors. In the future directions section, I indicated plans to incorporate additional variables in subsequent studies for a more comprehensive analysis of users' information-sharing behaviors. I believe these adjustments will strengthen the study and provide a more nuanced understanding of the complexities surrounding information-sharing behaviors during crises. Thank you for guiding me in this process.

R3: The paper relies on the cognitive appraisal theory of emotion (CATE) and construal level theory (CLT). While these theories have been discussed in many papers, the authors have to use new theories.

Authors’ Response: Thank you for your constructive feedback and encouragement. I greatly appreciate your valuable feedback on my research. Following your suggestion, I conducted further research to explore relevant theories that can explain information-sharing behaviors and identified the novel HSM as a more innovative framework.

I have revised the outdated CATE to the HSM and subsequently reconstructed the research model based on this theory. However, I would like to acknowledge that despite an extensive literature review, I was unable to find a newer theory to replace the CLT. At present, it seems that no emerging theory can fully supplant the CLT.

I hope you understand this and accept my sincere apology. I am grateful for your guidance in this study.

R4: The problem not about the methodology, the problem about the study model, it is weak model. It is crucial to include a dedicated section in the literature review that explains the foundational theories supporting the study. This section should delve into the theoretical frameworks that form the basis of the research, offering essential context for developing the subsequent model. the authors must to review more theories in the field to add new variables.

Authors’ Response: I want to express my gratitude for your valuable feedback on my research. Recognizing the limitations of the previous model in terms of explanatory power, I endeavored to construct a more robust research model. Building upon the introduced HSM and considering the crisis context central to this study, I have restructured the research model by delineating cognition and emotion along heuristic and systematic pathways.

Moreover, to provide a comprehensive understanding of the model's relevance to this study, I have introduced the model and its applicability in the introduction section. In section 2.2, I have further elaborated on the model by discussing the HSM in conjunction with the Elaboration Likelihood Model (ELM) from an information processing perspective. I hope that the newly constructed model will offer stronger support for this study.

I appreciate your insightful guidance and suggestions, and I hope that the enhancements made to the research model will contribute significantly to the study's findings. Thank you once again for your valuable input.

R5: The conclusion is missing.

Authors’ Response: Following the constructive feedback you provided, I have re-examined the research findings based on the modified theoretical foundation and research hypotheses. This has further effectively addressed the research question. With the updated model and research findings in mind, I have refined the study's conclusions to offer a more robust and comprehensive closing statement that encapsulates the key insights and implications of the research.

R6: The quality of Figure One is poor.

Authors’ Response: Thank you for your valuable suggestions and support. I have carefully addressed the concerns regarding the quality of Figure One. I am grateful for your guidance and input throughout this enhancement process.

Once again, I want to express my appreciation for your insightful feedback and constructive criticism, which have been invaluable in guiding the refinement of the paper. I would like to express my sincere appreciation for your guidance and insights throughout this process.

Round 3

Reviewer 2 Report

Comments and Suggestions for Authors

The paper is improved compared to the previous version, with enhancements in the theoretical section, the scientific contributions, and the justification for the study model.